# Isolation and Potential Biological Applications of Haloaryl Secondary Metabolites from Macroalgae

**DOI:** 10.3390/md17020073

**Published:** 2019-01-22

**Authors:** Ana Jesus, Marta Correia-da-Silva, Carlos Afonso, Madalena Pinto, Honorina Cidade

**Affiliations:** 1Laboratório de Química Orgânica e Farmacêutica, Departamento de Ciências Químicas, Faculdade de Farmácia, Universidade do Porto, Rua de Jorge Viterbo Ferreira 228, 4050-313 Porto, Portugal; anaaimjesus@gmail.com (A.J.); cafonso@ff.up.pt (C.A.); madalena@ff.up.pt (M.P.); 2Centro Interdisciplinar de Investigação Marinha e Ambiental (CIIMAR), Universidade do Porto, Terminal de Cruzeiros do Porto de Leixões, Avenida General Norton de Matos s/n, 4450-208 Matosinhos, Portugal

**Keywords:** macroalgae, secondary metabolites, haloaryl compounds, biological activity

## Abstract

Macroalgae have been reported as an important source of halogenated aromatic secondary metabolites, being the majority of these derivatives isolated from red algae. Halophenols and haloindoles are the most common haloaryl secondary metabolites isolated from these marine organisms. Nevertheless, some halogenated aromatic sesquiterpenes and naphthalene derivatives have also been isolated. Most of these secondary metabolites showed interesting biological activities, such as antitumor, antimicrobial, antidiabetic, and antioxidant. This review describes in a systematic way the distribution and natural occurrence of halogenated aromatic secondary metabolites from extracts of red, brown, and green algae, as well as biological activities reported for these compounds.

## 1. Introduction

The search for bioactive compounds from marine organisms in recent decades has produced an abundance of secondary metabolites with pharmaceutical and industrial applications. Among these marine natural products, the isolation of halogenated derivatives from macroalgae has been exhaustively reported. This work describes in a systematic way the distribution, natural occurrence, and biological activities of aromatic secondary metabolites with halogens on the aromatic moiety.

Over the past four decades, reports about the isolation of haloaryl secondary metabolites from macroalgae have been increasing, describing about two hundred halogenated aromatic secondary metabolites. Among algae, macroalgae—including brown, green, and red algae—are an important source of these secondary metabolites, with red algae being responsible for the production of nearly 90% of these compounds identified thus far (Figure 1).

In spite of the variety of macroalgae families, most of these derivatives have been isolated from the Rhodomelaceae family, especially *Laurencia* (54 compounds), *Rhodomela* (43 compounds), *Symphyocladia* (23 compounds), *Polysiphonia* (20 compounds), and *Odontthalia* (12 compounds) (Figure 2).

The haloaryl secondary metabolites containing bromine are more common (176 compounds) than with chlorine (14 compounds) and iodine (9 compounds) (Table 1). Interestingly, the number of secondary metabolites with chlorine is very similar to that with iodine, which would not be expected because chloride and bromide are much more abundant than iodide in seawater [1]. According to Lavoie et al. (2017), this disproportionately high number of iodinated compounds can be explained by the higher oxidation potential of iodide compared to bromide and chloride, allowing its faster oxidation by haloperoxidases, and their incorporation into the biosynthetic pathway of the secondary metabolites [2,3]. It is noteworthy that the halogenation degree is relatively higher for brominated metabolites than for chlorinated and iodinated metabolites (Table 1).

Considering the aryl scaffold, a variety of derivatives have been isolated, including halophenols (117 compounds), indoles (46 compounds), sesquiterpenes (18 compounds), and naphthalene derivatives (3 compounds) (Table 1).

Among halophenols, the isolation of monoaryl bromophenols (70 compounds) and bromophenol dimers with a methylene bridge between the two phenyl rings is quite usual (25 compounds), as well as the description of bromophenol dimers with different linkers such as oxygen (5 compounds), oxybis(methylene) (4 compounds), ethylene (4 compounds), and a carbonyl group (1 compound). The isolation of halophenol trimers (3 compounds) and tetramers (1 compound) has also been reported (Table 1).

Although halophenols have been isolated from a wide range of genera, the isolation of halo-indoles, -sesquiterpenes and -naphthalene derivatives have been mainly reported from a restricted number of genera, namely, the *Rhodophyllis* and *Laurencia* genera. 

Among haloindoles, the isolation of mono-indoles (39 compounds) is more common than dimers (only 7 compounds) and, in contrast to halophenols, the isolation of trimers or tetramers of haloindoles was not described (Table 1). Concerning the nature of the halogen, there are many more cases of indoles with chloro (13 compounds) or iodo (6 compounds) than in the halophenols class (Table 1). Nevertheless, the majority of the indoles presents two or three bromine atoms as happens with the phenol class.

Among halosesquiterpenes, the isolation of monoaryl sesquiterpenes (17 compounds) is more usual than dimers (only one example was found) and again the isolation of trimers or tetramers was not described (Table 1). Concerning the halogen, only bromosesquiterpenes have been found until now.

The naphthalene class is restricted to three bromonaphthalene derivatives.

The structure, natural occurrence, and biological activities of haloaryl secondary metabolites isolated from macroalgae are presented in alphabetical order by clade and genus in the next sections. Further information is provided in Appendix A.

## 2. Haloaryl Secondary Metabolites Isolated from Macroalgae

### 2.1. Haloaryl Secondary Metabolites Isolated from Red Algae

Red algae are probably one of the oldest groups of eukaryotic algae with a high diversity of families and genera, being one of the richest sources of bioactive secondary metabolites [4]. A total of 167 haloaryl secondary metabolites, including indoles, halophenols, and aromatic sesquiterpenes belonging to the Cystocloniaceae, Halymeniaceae, Lithothamniaceae, and Rhodomelaceae families, were isolated from this clade (Appendix A).

#### 2.1.1. Cystocloniaceae Family

Recently, 14 polyhalogenated indoles (**1**–**14**) were isolated from *Rhodophyllis membranaceae* and evaluated for cytotoxic and antifungal activities (Figure 3, Appendix A) [5]. In addition to indoles with only bromine substituents (compounds **1** and **2**), indoles with both bromine and chlorine (compounds **3–8**) and indoles with chlorine and iodine (compounds **9** and **10**) have also been isolated. Unusual indoles with the exceptionally rare presence of bromine, chlorine, and iodine were also found (compounds **11**–**14**). Compounds **1**, **3**, **4**, **6**, and **11** revealed interesting cytotoxic activity in the acute promyelocytic leukemia (HL-60) cell line (IC_50_ values between 28 and 78 µM) and antifungal activity against *Saccharomyces cerevisiae* (IC_50_ values between 23 and 83 µM) with compound **6**, with bromine and chloride at position 5 and 3, respectively, exhibiting the best results [5]. Changing the bromine to position 3 and chloride to position 5 led to a non-active compound.

#### 2.1.2. Halymeniaceae Family

As a result of the search for new α-glucosidase inhibitors with antidiabetic activity by Kim et al., two bromophenols isolated from *Grateloupia elliptica* (compounds **15** and **16**) [6] and one bis-bromophenol ether (BDDE, **17**) isolated from *Polyopes lancifolius* [7] were identified (Figure 3, Appendix A). Among these, BDDE (**17**) revealed the most potent activity, showing IC_50_ values of 0.098 μM and 0.120 μM against *Saccharomyces cerevisiae* and *Bacillus stearothermophilus* α-glucosidase, respectively, and 1.00 mM and 1.20 mM against rat intestinal sucrase and maltase [7]. Moreover, while compounds **15** and **16** showed a mixed type of inhibition against *S. cerevisiae* α-glucosidase, compound **17** displayed a competitive mixed type of inhibition [6,7]. The results obtained for these compounds suggest their potential application as nutraceuticals for the management of type 2 diabetes. BDDE (**17**) has also been isolated from other red macroalgae of the family Rhodomelaceae, namely, *Odonthalia corymbifera* [8] and *Rhodomela confervoides* [9], as well as from the brown algae *Leathesia nana* (Chordariaceae) [10], being also known for the promising antioxidant [11], antidiabetic [7,12], antitumor [10,13], antifungal [14], and antibacterial activities [15].

Liu et al. demonstrated that BDDE (**17**) has antifungal activity against several phytopatogenic fungi, namely, *Valsa mali*, *Fusarium graminearum*, *Coniothyrium diplodiella*, *Colletotrichum gloeosporioides*, and *Botrytis cinerea* [14]. Further studies revealed that **17** caused the disruption of the cell membrane integrity in *Botrytis cinerea* spores and newly formed germ tubes, as well as interacted with DNA via intercalation and minor groove binding [14]. These studies provided evidence that BDDE (**17**) has potential application in the control of gray mold after fruit harvest and could serve as a lead for the rational drug design of new antifungal agents [14].

This marine bromophenol (**17**) also has cytotoxic activity against several human tumor cell lines, through the interference with different cellular and molecular targets [10,16]. Liu et al. reported that compound **17** exhibited promising apoptotic activity in K562 cells via mitochondrial pathway and inhibited the activity of topoisomerase I, this effect being associated with the binding to the DNA minor groove [16]. Moreover, it was demonstrated that BDDE (**17**) acts as a potent antiangiogenesis agent both in vitro and in vivo [13]. In fact, this compound displayed in vitro antiangiogenesis ability through the inhibition of HUVEC cell proliferation, migration, and tube formation, and blocked in vivo subintestinal vessel formation in zebrafish embryos [13].

#### 2.1.3. Lithothamniaceae Family

Only one halophenol is known from this family. Lithothamnin A (**18**), the first bastadin-like analogue isolated from a red alga, was described in 2011 from *Lithothamnion fragilissimum* collected east of Lighthouse Reef, Palau Island (Figure 3, Appendix A) [17]. Compound **18**, with five bromines, exhibited modest antiproliferative activity against several human tumor cell lines, including melanoma (LOX), astrocytoma (SNB-19), ovarian serous adenocarcinoma (OVCAR-3), colon adernocarcinoma (COLO-205), and acute lymphyoblastic leukemia (MOLT-4) cell lines, showing IC_50_ values between 7.6 and 19.0 µM [17].

#### 2.1.4. Rhodomelaceae Family

Among all red algae, the Rhodomelaceae family is the main producer of haloaryl derivatives with a total of 150 secondary metabolites isolated from *Callophycus*, *Laurencia*, *Odonthalia*, *Osmundaria*, *Polysiphonia*, *Rhodomela*, *Symphyocladia*, and *Vidalia* species.

##### *Callophycus* Genus

In 2017, four new iodinated and brominated meroditerpenes (iodocallophycols A to D, **19–22**) with a unique structure were discovered from the Fijian red alga *Callophycus* sp. by Lavoie et al. (Figure 3, Appendix A) [3]. At 10 µM, none of the compounds revealed antibiotic activity against several wild-type and resistant bacterial strains [3].

##### *Laurencia* Genus

*Laurencia*, which is widely distributed along the coast in tropical and subtropical areas around the world, is one of the most important sources of bioactive haloaryl secondary metabolites from Rhodomelaceae, with approximately 54 compounds of this family being reported (**23**–**76**, Figure 4, Appendix A), including mainly haloindoles and aromatic sesquiterpenes. 

Bromo-1*H*-indole derivatives **23**–**26** were originally isolated from *Laurencia brongniartii* by Carter et al., in 1978, during an expedition in the Caribbean Sea [18]. Further reports include the isolation of these compounds from other species of *Laurencia*, namely, compounds **24** and **25** from *L. similis* [19], and **24**–**26** from *L. decumbens* [20] and *L. complanata* [21]. Bromo-1*H*-indoles **27**–**30** have also been reported from *Laurencia* sp., namely, *L. similis* [19] and *L. decumbens* [20]. Compounds **24** and **25** showed antimicrobial activity. Compound **24**, with three bromines, showed antibacterial activity against *Staphylococcus* sp. with a minimum inhibitory concentration (MIC) value of 300 µg/mL [22]. Compound **25**, with four bromines**,** displayed activity against *Bacillus cereus* and *B. subtilis*, *Saccharomyces cerevisiae, Staphylococcus aureus*, *Streptococcus pneumoniae*, and *Candida albicans* (Appendix A) [18,21]. In addition, this compound revealed antiproliferative activity in mouse lymphocytic leukemia tumor cells (L1210) (ID_50_ value of 3.6 µg/mL) [18]. In 1989, Tanaka et al. isolated 23 polybromoindoles (**23**–**26**, **31**–**49**) [23]. Itomanindole B (**32**) as well as the structure-related polybromoindoles **33** and **50**–**52** were also isolated from the same species by El-Gamal et al. [24] and evaluated for their cytotoxicity against human colon adernocarcinoma (HT-29) and mouse lymphocytic leukemia (P-388) cell lines. Only bisindoles **51** and **52**, both with sulfoxide groups, revealed cytotoxicity against the human tumor cell lines. In particular, compound **51** showed activity in both HT-29 and P-388 cell lines, and bisindole **52** exhibited cytotoxicity against the P-388 cell line [24]. The bisindole **53** was also isolated from *L. similis* [25] algae.

Red algae of the genus *Laurencia* are a rich source of halogenated aromatic sesquiterpenes, including secondary metabolites **54**–**71** isolated from several species of *Laurencia* [22,26,27,28,29]. Among these compounds, cupalaurenol (**54**) showed antibacterial activity against *Staphylococcus aureus*, *Staphylococcus* sp., *Salmonella* sp., and *Vibrio cholerae*, with MIC values between 125 and 200 µg/mL [22]. Laurinterol (**55**), bromolaurenisol (**67**), and the dimeric sesquiterpene **70**, biogenetically derived by ortho coupling of two laurinterol molecules, displayed moderate cytotoxic activity against several human tumor cell lines [27].

In addition to haloindoles and aromatic sesquiterpenes, *Laurentia* sp. have also been an interesting source of other bioactive halogenated aromatic secondary metabolites, namely, naphthalene, benzophenone, and diphenyl ether derivatives. Highly brominated secondary metabolites **72–76** were isolated from red alga *L. similis* and evaluated for their inhibitory activity against protein tyrosine phosphatase 1B (PTP1B). All compounds displayed inhibitory effect in this enzyme, compounds **75** and **76** being the most potent, with IC_50_ values of 2.66 µg/mL and 2.97 µg/mL, respectively [30].

##### *Odonthalia* Genus

From the *Odonthalia* genus, only bromophenols have been isolated. From these secondary metabolites, three are monoaryl bromophenols and six are bis-bromophenols.

Kurihara et al. identified five bromophenols (**17**, **77**–**80**) including BDDE (**17**) and lanosol (**77**) from the red alga *Odonthalia corymbifera* (Figure 5, Appendix A) [8]. Compounds **17**, **77**, and **78** behave as yeast α-glucosidase inhibitors (**17**: IC_50_ = 0.098 µM, **77**: IC_50_ = 89 µM, and **78**: IC_50_ = 25 µM), the symmetrical dimer BDDE (**17**) being the most potent [8]. As these compounds act as irreversible inhibitors, it was proposed that the α-glucosidase inhibition might result from the interaction of *o*-quinones which are oxidative products of *o*-diphenols, such as **17**, to enzyme protein. Moreover, the same research group studied the effect of these bromophenols as well as structure-related compounds in yeast and rat intestinal α-glucosidase sucrase and maltase [31]. Bromophenols **17**, **77**, and **78** displayed activity for both sucrase and maltase enzymes, with IC_50_ values between 1.1 and 3.5 mM [31].

More recently, from the same species of red alga collected along the coast of Sokcho, South Korea, six bromophenols (**77** and **81**–**85**) were isolated and evaluated for their effect as isocitrate lyase (ICL) inhibitors, a key enzyme in the glyoxylate cycle highly expressed during appressorium-mediated plant infection by the fungal pathogen of rice *Magnaporthe grisea* (Figure 5, Appendix A) [32]. All compounds were ICL inhibitors, being bromophenols **83** (IC_50_ = 2.1 ± 0.1 µM), **84** (IC_50_ = 2.8 ± 0.2 µM), and **85** (IC_50_ = 2.0 ± 0.1 µM) more potent than 3-nitropropionate, a well-known ICL inhibitor used as positive control. Interestingly, biarylbromophenols **83**–**85** displayed stronger ICL inhibitory activity than the simple brominated phenols such as **77**, **81**, and **82**, and the debromination of all compounds resulted in the loss of the inhibitory effect upon ICL activity. Collectively, these data indicate that the diphenylmethane skeleton and bromine moiety of bromophenols are essential for potent inhibition of ICL activity [32].

Islam et al. isolated two new bromophenols with a unique structure, odonthadione (**86**) and odonthalol (**87**), from the alga *Odonthalia corymbifera* (Figure 5, Appendix A) [33]. Odonthadione (**86**) is a hybrid of a brominated hydroxylated benzyl (BHB) unit and a cyclopentenedione moiety, and odonthalol (**87**) is a BHB unit trimer with an ether linkage. The isolated algal bromophenols **86** and **87** were investigated for antioxidant and tyrosinase inhibitory activities [33]. Both compounds revealed DPPH and ABTS radical scavenging activity, showing EC_50_ values between 24.7 µM and 6.7 µM [33]. Compound **86** displayed two-fold stronger tyrosinase inhibitory activities than kojic acid, used as the positive control, whereas compound **87** showed only a slightly higher activity [33].

##### *Osmundaria* Genus

Popplewell and Northcote reported the isolation of a new nitrogenous bromophenol, colensolide A (**88**), together with the known bromophenol lanosol (**77**), as well as its methyl ether (**81**), and the aldehyde (**89**) and butenone (**90**) derivatives of lanosol and rhodomelol (**91**) from the New Zealand red alga *Osmundaria colensoi* (Figure 5, Appendix A) [34]. Compounds **77**, **81**, and **88**–**91** were evaluated for cytotoxicity against the HL-60 human leukemia cell line and for antibacterial activity against the MC2155 strain of *Mycobacterium smegmatis*. Lanosol butenone (**90**) exhibited moderate activity against HL-60 human leukemia cells (IC_50_ = 8.0 µM), whereas lanosol methyl ether (**81**), lanosol butenone (**90**), and rhodomelol (**91**) exhibited antibacterial activity against the MC2155 strain of *Mycobacterium smegmatis* (IC_50_ values of 7.8, 26.2, and 28.1 µM, respectively) [34].

##### *Polysiphonia* Genus

Two bromophenols (**92**, **93**) were isolated from *Polysiphonia morrowii*, collected in Hakodate, southern Hokkaido, Japan (Figure 5, Appendix A). Compound **92** exhibited some effect as both yeast and rat intestinal sucrase and maltase α-glucosidase inhibitors (yeast α-glucosidase: IC_50_ = 100 µM; rat intestinal sucrose α-glucosidase: IC_50_ = 3.6 mM; rat intestinal maltase α-glucosidase: IC_50_ = 4.8 mM) [31].

Shoeib et al. identified by gas-liquid chromatography-mass spectrometry (GLC-MS) analysis, lanosol (**77**) and the methyl, ethyl, and n-propyl ethers of lanosol (**81**, **94**, and **95**, respectively), as well as the aldehyde of lanosol (**89**), in the chloroform fraction of red alga *Polysiphonia lanosa*, and all compounds showed in vitro cytotoxic activities against human colon cell lines DLD-1 and HCT-116 cells (Figure 5, Appendix A) [35].

Seven new natural occurring bromophenols **96**–**102** as well as known compounds **103**–**106** were identified from the marine red alga *Polysiphonia urceolata* (Rhodomelaceae) and evaluated for their DPPH radical scavenging activity (Figure 5, Appendix A) [36,37,38]. All compounds revealed to be more potent than butylated hydroxytoluene (BHT), a well-known antioxidant agent used as the positive control, showing IC_50_ values ranging from 6.1 to 35.8 µM [36,37,38]. Among these compounds, some revealed a unique structure, particularly 9,10-dihydrophenantrenes (**99**,**100**), 5,7-dihydrodibenzo[c,e]oxepine (**101**), and urceolatin (**102**). Bromophenols **99** and **100** represent the second example of 9,10-dihydrophenantrenes isolated from marine sources, the first example reported being (±) – polysiphenol (**107**) isolated in 1990 from the red alga *Polysiphonia ferulacea* collected at Joal, Senegal [39]. It has been proposed that these compounds may biosynthetically derive from dihydrostilbene derivatives by oxidative phenolic coupling [40]. 

As a result of the bioguided fractionation of red alga *P. morrowii*, compound **106** as well as the structure-related simple bromophenol **108** were identified as promising antiviral compounds against two fish pathogenic viruses, namely, infectious hematopoietic virus (IHNV) and infectious pancreatic necrosis virus (IPNV) (Figure 5, Appendix A) [41]. For both compounds, the concentration causing a 50% inhibition of flounder spleen cell (FSP cell) proliferation (CC_50_) and each viral replication (EC_50_) were measured. Both compounds exhibited antiviral activity with selective index (CC_50_/EC_50_) values of 20 and 42 against IHNV and IPNV, respectively. These results suggest the possible application of these compounds on the discovery of new beneficial agents against viral diseases of salmonid fish, which causes serious losses to the trout and salmon industries [41].

##### *Rhodomela* Genus

*Rhodomela confervoides*, an alga commonly found along the coastlines of China, Japan, and Korea, has been reported as a source of bromophenols with diverse pharmacological activities, such as antibacterial, antitumor, antidiabetic, and antioxidant (Appendix A). In 2003, bis-phenols **109** and **110** (Figure 6), described for the first time, as well as the known bis-phenols **17**, **83**, and **85** were isolated from the methanolic extract of this marine alga and evaluated for their antibacterial activity against four ATCC standard bacteria strains (*Staphylcoccus aureus* ATCC29213, *Staphylcoccus epidermidis* ATCC12228, *Escherichia coli* ATCC25922, and *Pseudomonas aeruginosa* ATCC27853), as well as four bacteria strains isolated from clinic (*S. aureus* 02-60, *S. epidermidis* 02-04, *E. coli* 02-26, and *P. aeruginosa* 02-29). All compounds exhibited antibacterial activity, with compound **17**, the only bis-phenol linked by an oxygen atom, being the most active [15]. The same researchers isolated metabolite **111**, a bromophenol derivative with an aliphatic chain as substituent, found in *R. confervoides* algae extracts [9]. Later, the promising antibacterial activity on these bacterial strains by bromophenol **17** was confirmed by Han et al. [42]. In addition to this compound, these researchers reported the isolation of the new monoaryl bromophenol **112** and two known structure-related secondary metabolites **113** and **114** (Figure 6). Compounds **17** and **112**–**114** showed antiproliferative activity in several human tumor cell lines (epithelial tumor cell (KB), human hepatocellular carcinoma (Be17402), and lung cancer cells (A549)), with **112**, a bromophenol with an ester group, being the most potent (3.54 < IC_50_ < 3.09 µg/mL) [42]. Ma et al. reported the isolation of eight new bromoaryl secondary metabolites with an unusual structure, particularly bromophenols with a C–N coupled with methyl γ-ureidobutyrate (**115**–**118**), the phenylethanol bromophenol (**119**), and three phenylethanol sulfate bromophenols (**120**–**122**) from *R. confervoides* (Figure 6) [43]. Among these secondary metabolites, only halophenols **119**–**122** displayed moderate cytotoxicity against a panel of five human cancer cell lines—lung adenocarcinoma (A549), human ovarian (A2780), hepatoma (Bel7402), stomach (BGC-823), and human colon (HCT-8) cancer cell lines [43].

Protein tyrosine phosphatase 1B (PTP1B) acts as a negative regulator in insulin signaling pathways, therefore being an effective target for the treatment of type 2 diabetes mellitus. Considering the importance of this phosphatase for diabetes treatment, the PTP1B inhibitory effect of bromophenols **17**, **85**, **110**, and **123** (Figure 6) isolated from *R. confervoides* was screened [9,12]. All compounds revealed potent inhibitory effect (2.4 < IC_50_ < 0.84 µM), with BPN (**110**) being particularly active (IC_50_ = 0.84 µM) [12].

In addition to the PTP1B inhibitory effect of BDDPM (**85**), this secondary metabolite has been described as a potent antitumor agent, having inhibitory effect on a wide range of human tumor cell lines—cervical (Hela), colon (RKO and HCT-116), hepatoma (Bel-7402), vascular endothelial (HUVEC), and glioblastoma (U87) cancer cells [44].

Bromophenols, including monoaryl **77**, **81**, **89**, **90**, **94**, **119**, and **124**–**135**, diaryl **17**, **83**–**85**, **110**, and **136**, and triaryl derivatives **123** as well as seven new nitrogen-containing bromophenols (**137–143**) were isolated from the methanolic and ethyl acetate extracts of *R. confervoides* and tested for their antioxidant potential using the DPPH and ABTS scavenging activity assay (Figure 6) [11,45]. All compounds showed potent scavenging properties for both radicals, exhibiting similar or even lower IC_50_ values than BHT and ascorbic acid, two well-known antioxidants used as positive controls. Compounds **77** (lanosol), **83**, and **84** presented also DPPH and ABTS radical scavenging activity (**77**: IC_50_ (DPPH) = 9.52 ± 0.04 µM, TEAC (ABTS) = 2.06 ± 0.11 nM; **83**: IC_50_ (DPPH) = 14.32 ± 0.12 µM, TEAC (ABTS) = 3.00 ± 0.13 nM), and **84**: IC_50_ (DPPH) = 19.60 µM, TEAC (ABTS) = 3.16±0.14 nM) [11]. These results suggest the potential of this marine alga as a source of antioxidants, which may be used to prevent the oxidative deterioration of food and as a nutritional supplement [11,45].

##### *Symphyocladia* Genus

*Symphyocladia latiuscula* (Harvey) Yamada is a member of the family Rhodomelaceae widely distributed along the coasts of northern China, Korea, and Japan [46]. This red alga is an important source of chemical diverse bromophenols, including monoaryl and diaryl secondary metabolites with antidiabetic, antioxidant, antifungal, and DNA polymerase inhibitory activities (Appendix A).

In 1999, Kurihara et al. isolated compounds **144** and **145** (Figure 7) from this alga and tested their α-glucosidase inhibitory activity [31]. Both compounds proved to be promising yeast α-glucosidase inhibitors, showing IC_50_ values of 11 µM and 0.030 µM, respectively. This effect was also observed using rat intestinal sucrase and maltase (Appendix A) [31].

As a result of the search for new bioactive secondary metabolites from *S. latiuscula* with potential effects on diabetes, a total of nine bromophenols, including four diaryl derivatives **145**–**148** and five monoaryl derivatives **149**–**153** were isolated (Figure 7) [47,48]. Compounds **145**–**147**, **150**, and **151** displayed a significant aldose reductase inhibitory activity (0.11 < IC_50_ < 1.15 µg/mL), showing that all compounds had a higher inhibitory effect than the positive control quercetin (IC_50_ = 1.05 µg/mL), with the exception of compound **150**, which exhibited a similar activity [47]. Bromophenols **81**, **145**, **148**, **149**, **152**, and **153** revealed that they inhibited the PTP1B enzyme, with **145**, **148**, and **149** showing strong activity with IC_50_ values of 4.3, 3.5, and 3.9 μmol/L, respectively [48].

Bromophenols **144**, **145**, **148**–**151**, and **153**–**157** exhibited DPPH radical scavenging activity, with this effect being more potent than quercetin [49] or BHT [50] that were used as positive controls (Figure 7). Other compounds with this activity include two mono arylphenols possessing a methyl γ-ureidobutyrate (**158**) or *cis*-aconitic acid methyl ester (**159**) moieties (Figure 7) [46].

The antifungal activity of diaryl bromophenols **156**, **160**–**162** and bromobenzyl methyl sulfoxide **163** was studied by Xu et al., showing compounds **156**, **162**, and **163** a moderate growth inhibitory effect against *Candida albicans*, with MIC values ranging from 37.5 to 12.5 µg/mL (Figure 7) [51,52].

Monoaryl bromophenols SL-1 (**164**) and SL-2 (**165**) were tested as DNA polymerase inhibitors using the polymerase chain reaction assay (PCR) [53]. Jin et al. verified that 0.5 µg of SL-1 (**164**) inhibited the enzyme, whereas SL-2 (**165**) only had the capability to inhibit this enzyme at 5 µg (Figure 7) [53].

##### *Vidalia* Genus

The only report about haloaryl secondary metabolites described in algae from *Vidalia* sp. concerns the isolation of two bromophenols, vidalols A (**166**) and B (**167**), from the Caribbean red alga *Vidalia obtusaloba* (Figure 7) [54]. Wiemer et al. described that these two compounds significantly reduced the edema when applied topically to phorbol ester (PMA)-induced swelling of the mouse ear [54]. Moreover, both compounds inhibited bee venom phospholipase A_2_ (PLA_2_), showing an inhibition percentage of 96% at 1.6 µg/mL, suggesting their potential as lead compounds to design new PLA_2_ inhibitors [54]. According to Wiemer et al., the production of these bromophenols in *V. obtusaloba* could be important as a defense mechanism against some marine herbivores, an example being vidalol A (**166**) that has been shown to reduce the grazing of *Thalassia testudinum* by Caribbean herbivorous fishes [54,55].

### 2.2. Haloaryl Secondary Metabolites Isolated from Brown Algae

Brown seaweeds exhibit significant morphological diversity and are dominant in marine littoral zones from subpolar to equatorial regions. From algae of the Chordariaceae and Dictyotaceae families, 10 dimeric halophenols have been isolated.

#### 2.2.1. Chordariaceae Family

In 2004, Xu et al. isolated the dibenzyl bis-bromophenols **168**–**173** with different dimerization patterns and two propyl bromophenol derivatives (**174** and **175**), together with 11 known bromophenol derivatives (**17**, **77**, **81**, **83**, **85**, **89**, **94**, **110**, and **176**–**178**) from the ethanolic extract of the brown algae *Leathesia nana* (Figure 8). Among the isolated compounds, **83**, **85**, **110**, and **170** revealed potent cytotoxic effect against human cancer cell lines, especially lung adenocarcinoma (A549), stomach (BGC-823), breast (MCF-7), hepatoma (Bel7402), and human colon (HCT-8) cell lines, with IC_50_ values between 0.0018 and 0.0214 µM/mL [10].

Compounds **110**, **170**, and **172** exhibited potent in vitro growth inhibitory activity against eight human cancer cell lines (A549, BGC-823, MCF-7, B16-BL6, HT-1080, A2780, Bel7402, and HCT-8) with an IC_50_ value below 10 μg/mL, this effect being associated with a moderate inhibitory activity against protein tyrosine kinase (PTK) with over-expression of c-kit. Together, these results indicated that these bromophenol derivatives can be used as potent antitumor agents for PTK over-expression of c-kit [56].

#### 2.2.2. Dictyotaceae Family

In 2009, Areche et al. reported the isolation of the unusual chlorinated meroditerpenoid 4´-chlorostypotriol triacetate (**179**) (Figure 8) from the dichloromethane extract of the brown alga *Stypopodium flabelliforme* (Dictyotaceae family) collected in Easter Island, Chile. This compound was the first metabolite reported from the *Stypopodium* genus possessing one halogen atom [57].

### 2.3. Haloaryl Secondary Metabolites Isolated from Green Algae

The isolation of haloaryl secondary metabolites from green algae is uncommon, with a description of only five compounds isolated from the Cladophoraceae and Dichotomosiphonaceae families.

#### 2.3.1. Cladophoraceae Family

The new bromoindole **180** possessing both chlorine and bromine was isolated from the ethanolic extract of the green alga *Chaetomorpha basiretorsa* (Cladophoraceae) in 2005 (Figure 8) [58].

#### 2.3.2. Dichotomosiphonaceae Family

A total of four bromophenols were isolated from *Avrainvillea* sp., including the brominated diphenylmethanes **181–182**, the monoaryl phenol **183**, and the tetraarylphenol **184** (Figure 8). According to Carte et al., avrainvilleol (**181**) was identified from the ether extract of *A. longicaulis,* whereas its methyl ether was isolated from the methanol extract of the same species [59]. Studies of *A. nigricans* resulted in the isolation of not only avrainvilleol (**181**), but also the structure-related diaryl bromophenol **182**, and the monoaryl phenol **183** (Figure 8). Compounds **181–183** showed inhibitory activity against *Bacillus subtilis* and *Staphylococcus aureus*, with **183** being also active against *Pseudomonas aeruginosa*, *Escherichia coli*, *Serratia marcesens*, and *Candida albicans* [60]. In addition, bromophenol **183** showed to be an in vitro growth inhibitor of the human KB cancer cell line with an ED_50_ value of 8.9 µg/mL [60]. The secondary metabolite **184** exhibited HMG-CoA reductase inhibitory activity with an IC_50_ value of 5 µM (Figure 8) [59].

## 3. Conclusions and Perspectives

Marine macroalgae play an essential role in the marine environment for the production of oxygen and as a source of food for marine animals. Moreover, these organisms generate compounds and products utilized in many commercial fields, such as fertilizers, and help to obtain compounds with pharmaceutical, cosmetic, and industrial applications.

This review provides an overview of the most relevant haloaryl secondary metabolites isolated from macroalgae, including their distribution and biological activities. A total of 184 haloaryl secondary metabolites, including halophenols, indoles, aromatic sesquiterpenes, and naphthalene derivatives were isolated from macroalgae, with red algae currently being the most prominent source of these compounds, particularly several species of algae from the Rhodomelaceae family. Nevertheless, further biochemical analyses on green and brown macroalgae in the future may also result in the discovery of new compounds from other clades. Most of these halogenated compounds are brominated with a diverse degree of halogenation, as well as some examples of secondary metabolites with chlorine and iodine being described. The most abundant haloaryl derivatives are bromophenols, with most of them possessing at least one catechol group.

The biological potential of the majority of haloaryl secondary metabolites has been exhaustively reported, as they are well known their antioxidant, antitumor, antimicrobial, and antidiabetic activities. Therefore, it is expected that some of these compounds may be used in the future in drug discovery. As the distribution of many of the macroalgae is rare in nature, strategies for securing the sustainable production of these secondary metabolites must be implemented. One strategy for overcoming this bottleneck is by using bioprocess technology to produce cell and tissue cultures of marine macroalgae. In fact, the bioprocess engineering of macroalgae for the production of secondary metabolites has been an emerging area of marine biotechnology. Several cell and tissue cultures derived from marine macroalgae have been developed, not only to facilitate the study of secondary metabolites biosynthesis, but also to allow the manipulation and controlled production of these compounds [61,62,63]. Other strategies may include the chemical synthesis of these or nature-inspired haloaryl compounds.

Among the bioactive compounds, bromophenols possessing 2,3-dibromo-3,4-dihydroxy phenyl rings, such as BDDE (**17**) and BDDPM (**85**), are the most promising. In fact, both compounds revealed to be quite active in a diverse array of biological activities, especially antitumor and antidiabetic. Taking these results into account, it will be interesting to develop new BDDE and BDDPM synthetic analogues in order to explore the potential of these compounds as leads for drug discovery.

Although several studies about the biological potential of these macroalgae natural products have been described, some unique indoles and aromatic sesquiterpenes have not been explored concerning their biological potential. Therefore, it is expected that the future exploitation of these haloaryl derivatives may contribute to medicinal chemistry in the discovery of innovative bioactive compounds.

## Figures and Tables

**Figure 1 marinedrugs-17-00073-f001:**
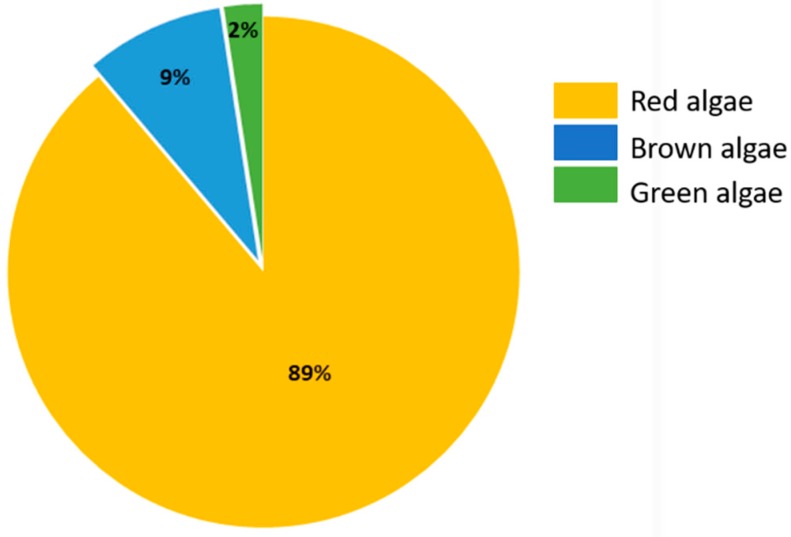
Distribution of the haloaryl secondary metabolites in macroalgae by clade.

**Figure 2 marinedrugs-17-00073-f002:**
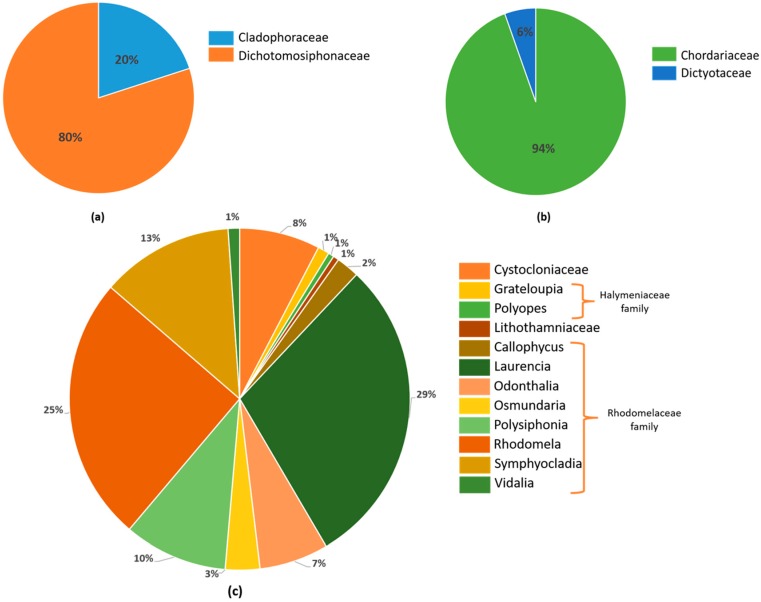
Distribution of the haloaryl secondary metabolites in (**a**) green, (**b**) brown, and (**c**) red algae by genus and family.

**Figure 3 marinedrugs-17-00073-f003:**
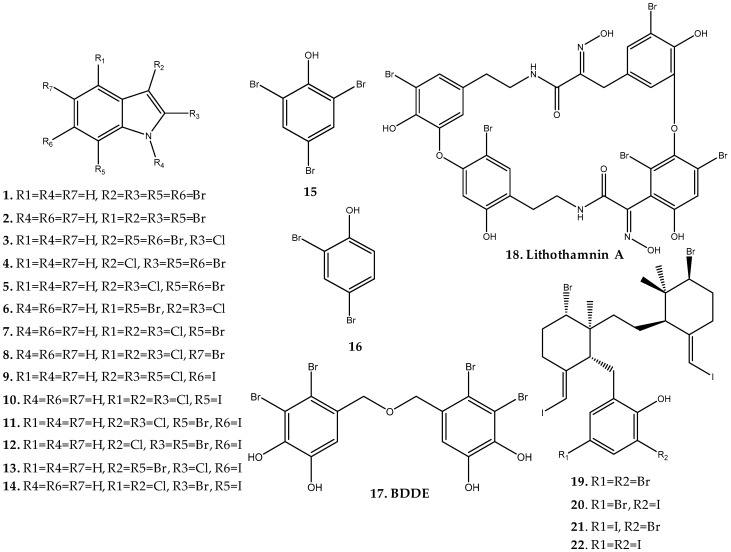
Naturally occurring haloaryl secondary metabolites **1**–**22**. Further information is provided in Appendix A.

**Figure 4 marinedrugs-17-00073-f004:**
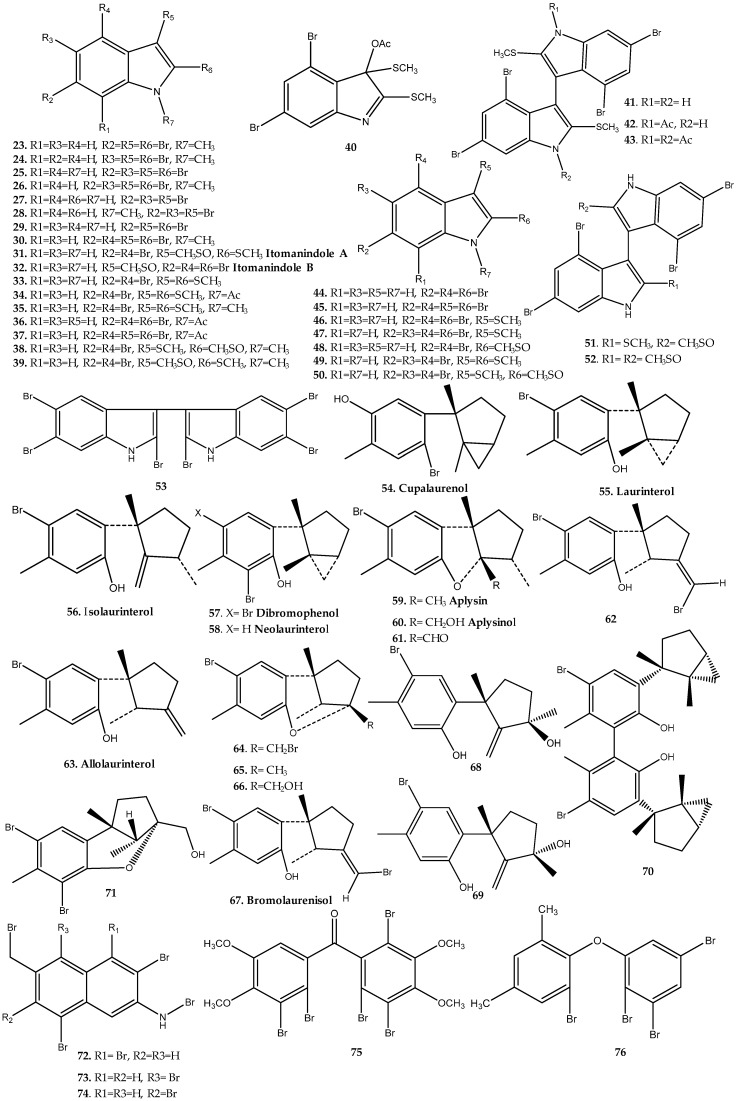
Naturally occurring haloaryl secondary metabolites **23**–**76**. Further information is provided in Appendix A.

**Figure 5 marinedrugs-17-00073-f005:**
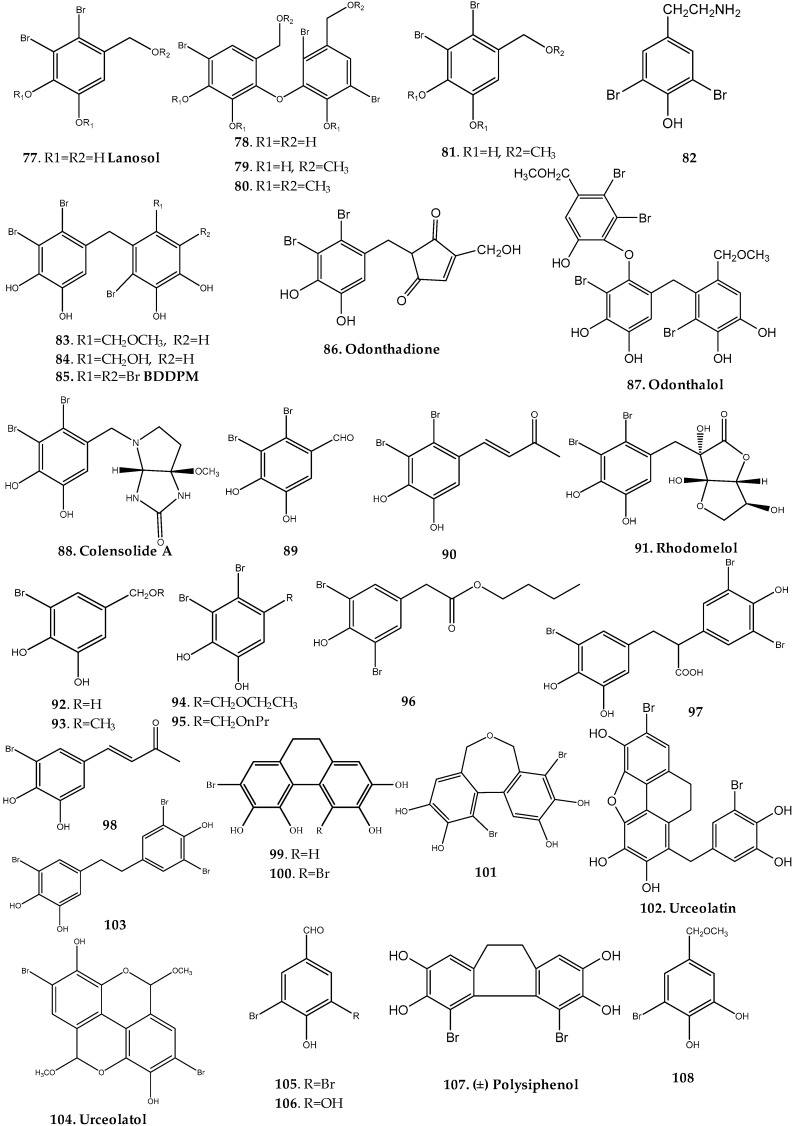
Naturally occurring haloaryl secondary metabolites **77**–**108**. Further information is provided in Appendix A.

**Figure 6 marinedrugs-17-00073-f006:**
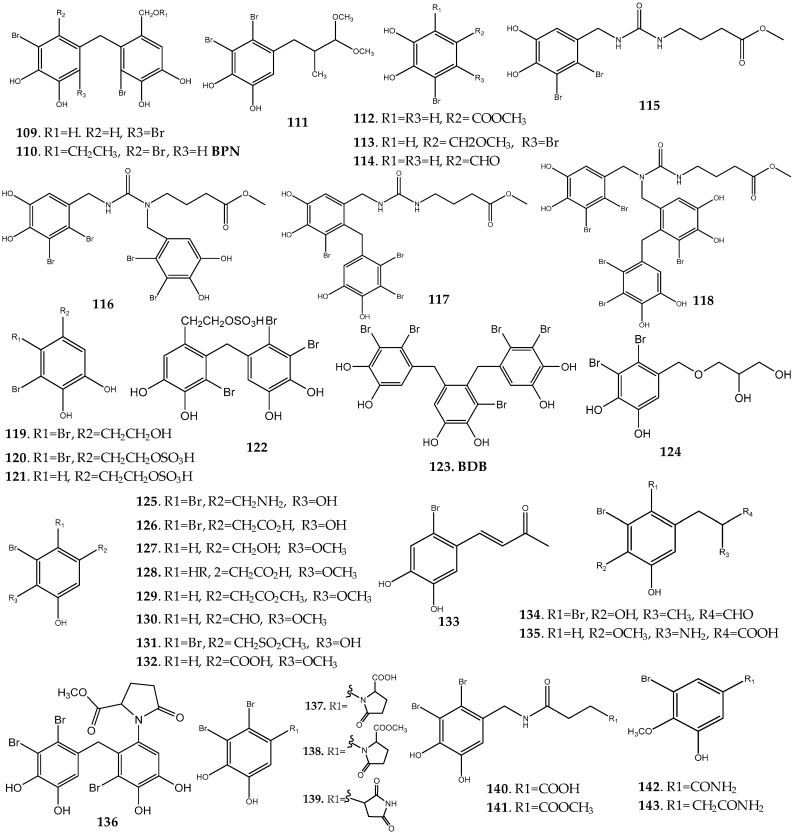
Naturally occurring haloaryl secondary metabolites **109**–**143**. Further information is provided in Appendix A.

**Figure 7 marinedrugs-17-00073-f007:**
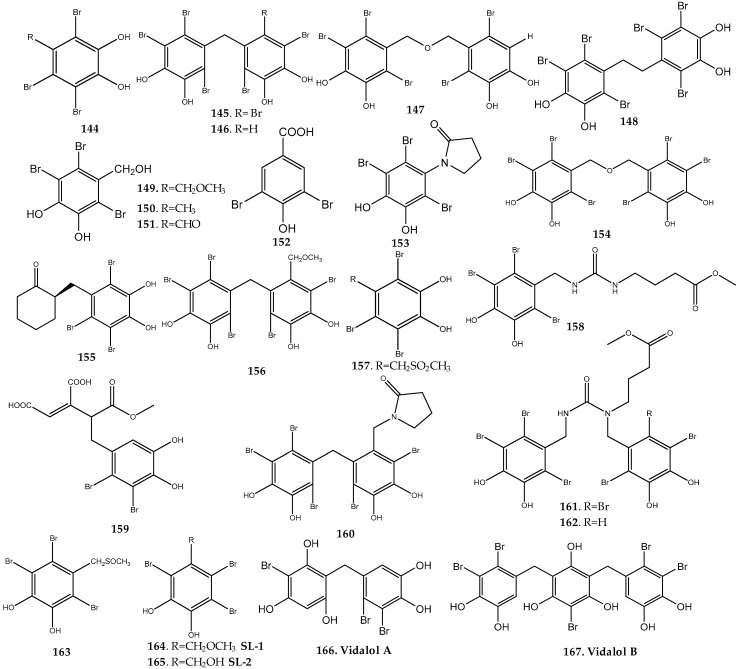
Naturally occurring haloaryl secondary metabolites **144**–**167**. Further information is provided in Appendix A.

**Figure 8 marinedrugs-17-00073-f008:**
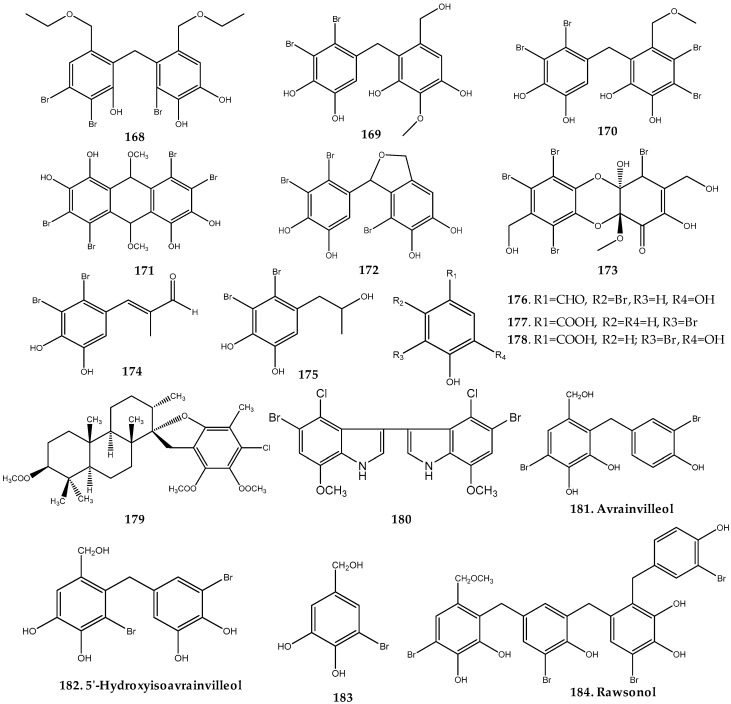
Naturally occurring haloaryl secondary metabolites **168–184**. Further information is provided in Appendix A.

**Table 1 marinedrugs-17-00073-t001:** Degree of halogenation of macroalgae haloaryl secondary metabolites.

	Number of Halogens																	
	Bromo						Chloro						Iodo					
	1	2	3	4	5	6	1	2	3	4	5	6	1	2	3	4	5	6
**Halophenols**	-	-	-	-	-	-	-	-	-	-	-	-	-	-	-	-	-	-
- monoaryl halophenols	23	35	12	-	-	-	1	-	-	-	-	-	2	1	-	-	-	-
- dimers	1	7	15	6	6	4	-	-	-	-	-	-	-	-	-	-	-	-
- trimers	-	-	-	1	2	-	-	-	-	-	-	-	-	-	-	-	-	-
- tetramers	-	-	-	1	-	-	-	-	-	-	-	-	-	-	-	-	-	-
**Indoles**	-	-	-	-	-	-	-	-	-	-	-	-	-	-	-	-	-	-
- monoaryl	4	11	13	8	-	-	3	5	4	-	-	-	6	-	-	-	-	-
- dimers	-	1	-	5	-	1	1	-	-	-	-	-	-	-	-	-	-	-
**Sesquiterpenes**	-	-	-	-	-	-	-	-	-	-	-	-	-	-	-	-	-	-
- monoaryl	15	2	-	-	-	-	-	-	-	-	-	-	-	-	-	-	-	-
- dimers	-	1	-	-	-	-	-	-	-	-	-	-	-	-	-	-	-	-
**Naphthalene derivatives**	-	-	3	-	-	-	-	-	-	-	-	-	-	-	-	-	-	-
Total of haloaryl derivatives	43	57	43	21	8	5	5	5	4	-	-	-	8	1	-	-	-	-

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
