# Peer review of "Isolation and Potential Biological Applications of Haloaryl Secondary Metabolites from Macroalgae"

_marinedrugs, 2019, doi:10.3390/md17020073_

Reviewer 1 Report

This manuscript represents a very useful review and catalog of haloaryl secondary compounds identified thus far from both current and older literature. Overall, this paper is a valuable resource and I recommend it for publication after minor revisions – mostly due to grammatical issues (see below).

One thing I feel needs to be improved is the authors’ “conclusion” that red algae are the main source of these compounds.  I will argue that this is true based on the current literature studies, but might change as more green and brown algae are analyzed for these types of compounds.  Therefore I think that lines 33 and 404 can be improved by softening the conclusions that red algae are “currently” the most prominent sources of these compounds, and that this might change as more biochemical analyses are performed in the future on algae.

Recommended Minor edits:

Line 32: "being red algae responsible for the production of nearly 90% of these compounds (Figure 1).” Change to: “with red algae being responsible for the production of nearly 90% of these compounds identified thus far (Figure 1).

Line 35 (and more (Line 85)): Please change the use of the word “phylo” I think it is not a word to be used.  I recommend the use of “clade” to replace “phylo” in all instances in this paper

Please designate Rhodomelaceae as a family: “…isolated from the rhodomelaceae family”

Line 48: change grammar to “the higher oxidation potential of iodide compared to bromide and chloride”

Line 62 – 64:  the plural form of genus is “genera”.  Please use “genera” in all three cases here.

Line 67: change to “…there are many more cases”

Line 118: change to “… also has cytotoxic…”

Line 189 (check line 377 also): change to “metabolites three are monoaryl bromophenols and six are bis-bromophenols.” For consistency with the rest of the manuscript

Line 355: change to “…families, 10 dimeric halophenols were isolated.”

Line 398-399 “…Moreover, these organisms generate compounds and products utilized in many commercial fields, namely as fertilizers…”

Line 404: “…derivatives were isolated from macroalgae, with red algae currently being the most prominent source of these compounds…”

Author Response

Response to Reviewer 1 Comments

This manuscript represents a very useful review and catalog of haloaryl secondary compounds identified thus far from both current and older literature. Overall, this paper is a valuable resource and I recommend it for publication after minor revisions – mostly due to grammatical issues (see below).

One thing I feel needs to be improved is the authors’ “conclusion” that red algae are the main source of these compounds. I will argue that this is true based on the current literature studies, but might change as more green and brown algae are analyzed for these types of compounds. Therefore I think that lines 33 and 404 can be improved by softening the conclusions that red algae are “currently” the most prominent sources of these compounds, and that this might change as more biochemical analyses are performed in the future on algae.

We understand the reviewer concern and we included that critical point of view in the conclusions, as suggested (lines 408-413).

Recommended Minor edits:

Line 32: "being red algae responsible for the production of nearly 90% of these compounds (Figure 1).” Change to: “with red algae being responsible for the production of nearly 90% of these compounds identified thus far (Figure 1).

We have changed line 32 as suggested

Line 35 (and more (Line 85)): Please change the use of the word “phylo” I think it is not a word to be used.  I recommend the use of “clade” to replace “phylo” in all instances in this paper

We have changed “phylo” to “clade” in all document

Please designate Rhodomelaceae as a family: “…isolated from the rhodomelaceae family”

We have designated accordingly

Line 48: change grammar to “the higher oxidation potential of iodide compared to bromide and chloride”

We thanks for the grammar correction

Line 62 – 64: the plural form of genus is “genera”.  Please use “genera” in all three cases here.

We thanks for the grammar correction

Line 67: change to “…there are many more cases”

We thanks for the grammar correction

Line 118: change to “… also has cytotoxic…”

We have changed as suggested

Line 189 (check line 377 also): change to “metabolites three are monoaryl bromophenols and six are bis-bromophenols.” For consistency with the rest of the manuscript

We have changed as suggested and reviewed all the document

Line 355: change to “…families, 10 dimeric halophenols were isolated.”

We have changed as suggested

Line 398-399 “…Moreover, these organisms generate compounds and products utilized in many commercial fields, namely as fertilizers…”

We have changed as suggested

Line 404: “…derivatives were isolated from macroalgae, with red algae currently being the most prominent source of these compounds…”      

We have changed as suggested

Reviewer 2 Report

This review provides an expansive update on the variety halogenated aromatic metabolites from macrophytic marine algae, with emphasis on bioactive metabolites.  The review provides a concise summary of the phyta where haloaryl metabolites are found, and their extents of halogenation.  In this regard, Table 2 is particularly valuable.   Below are two major points of consideration for a revised manuscript.

1. A major omission with this review is aryl monoterpenes from macrophytic red algae.   This reviewer does not understand why this important class was omitted, as aryl monoterpenes from algae may certainly exist.   Many review papers on this topic are available which the authors might closely analyze for relevance to this review.  If the authors still claim there are none, I hope they will provide documentation and fundamental perspectives towards this point.  

2. Given the emphasis in the “Conclusions and perspectives” section on potential uses of these metabolites, discussion on strategies for securing their sustainable production is warranted, as many of macrophytic algae bearing bioactive compounds are rarely distributed in nature.  Below are some references that may be of assistance in comments 1 and 2 above.

Rorrer, G.L., and Cheney, D.P.  “Bioprocess Engineering of Cell and Tissue Cultures for Marine Seaweeds.”  Aquacultural Engineering, 32, 11–41 (2004).

Polzin, J.J, and Rorrer, G.L., “Metabolic Flux Analysis of Halogenated Monoterpene Biosynthesis in Microplantlets of the Macrophytic Red Alga Ochtodes secundiramea.”  Biomolecular Engineering, 20, 205-215 (2003).

Barahona, L.F., and Rorrer, G.L.  “Isolation of Halogenated Monoterpenes from Bioreactor Cultured Microplantlets of the Macrophytic Red Algae Ochtodes secundiramea and Portieria hornemannii.”  Journal of Natural Products, 66, 743-751 (2003).

Author Response

Response to Reviewer 2 Comments

This review provides an expansive update on the variety halogenated aromatic metabolites from macrophytic marine algae, with emphasis on bioactive metabolites. The review provides a concise summary of the phyta where haloaryl metabolites are found, and their extents of halogenation. In this regard, Table 2 is particularly valuable. Below are two major points of consideration for a revised manuscript.

A major omission with this review is aryl monoterpenes from macrophytic red algae. This reviewer does not understand why this important class was omitted, as aryl monoterpenes from algae may certainly exist.   Many review papers on this topic are available which the authors might closely analyze for relevance to this review.  If the authors still claim there are none, I hope they will provide documentation and fundamental perspectives towards this point.  

Aryl monoterpenes are indeed an important class from macrophytic red algae. Nevertheless, this review includes in a systematic way the distribution, natural occurrence and biological activities of haloaryl compounds, i.e., aryl compounds bearing halogens directed linked to the aromatic moiety. We have now clarified this point in the beginning of the introduction (lines 28-30).

Given the emphasis in the “Conclusions and perspectives” section on potential uses of these metabolites, discussion on strategies for securing their sustainable production is warranted, as many of macrophytic algae bearing bioactive compounds are rarely distributed in nature.  Below are some references that may be of assistance in comments 1 and 2 above.

Rorrer, G.L., and Cheney, D.P.  “Bioprocess Engineering of Cell and Tissue Cultures for Marine Seaweeds.”  Aquacultural Engineering, 32, 11–41 (2004).

Polzin, J.J, and Rorrer, G.L., “Metabolic Flux Analysis of Halogenated Monoterpene Biosynthesis in Microplantlets of the Macrophytic Red Alga Ochtodes secundiramea.”  Biomolecular Engineering, 20, 205-215 (2003).

Barahona, L.F., and Rorrer, G.L.  “Isolation of Halogenated Monoterpenes from Bioreactor Cultured Microplantlets of the Macrophytic Red Algae Ochtodes secundiramea and Portieria hornemannii.”  Journal of Natural Products, 66, 743-751 (2003).

Strategies for securing sustainable production of macroalgae were included in the revised manuscript. All the suggested references were added. We thank the reviewer suggestion, which contributed for the enrichment of the “conclusions and perspectives” of this manuscript.